# Zinc Recovery from Wulagen Sulfide Flotation Plant Tail by Applying Ether Amine Organic Collectors

**DOI:** 10.3390/molecules26175365

**Published:** 2021-09-03

**Authors:** Zilong Ma, Lei Wang, Xiao Ni, Yinfei Liao, Zhian Liang

**Affiliations:** 1National Engineering Research Centre of Coal Preparation and Purification, China University of Mining and Technology, Xuzhou 221116, China; cumtmzl@126.com (Z.M.); ruiyin@126.com (Y.L.); 2Faculty of Materials Engineering, Baise University, Baise 533000, China; 3State Key Laboratory of Comprehensive Utilization of Low Grade Refractory Gold Ores, Shanghang 364200, China; zhian.liang@zijinmining.com; 4Mining and Metallurgy Research Institute, Zijin Mining Group Co., Ltd., Shanghang 364200, China

**Keywords:** froth flotation, smithsonite tailings, ether amine

## Abstract

Separating oxidized zinc minerals from flotation tailings is always a challenge. In this study, a flotation tailing from Wulagen zinc mine in China (Zn grade < 1%) was processed using froth flotation with combinations of amines (OPA 10, OPA 1214, OPA 13, DDA) and Na_2_S to study the effects of these amines on the zinc recovery as well as their interactions with other reagents, aiming to screen out a proper reagent scheme to improve zinc separation from extremely low-grade zinc flotation tailings. The results show that different amines led to different flotation performance, and the collectors were ranked as OPA 1214, OPA 13, OPA 10 and DDA in a decreasing order based on flotation collectivity and selectivity. An increase in the concentration of each collector increased the zinc recovery but reduced the concentrate zinc grade. Interactions were also observed between different amines and Na_2_S and Na_2_SiO_3_, and OPA 1214 outdid the others in saving the usage of both the Na_2_S and Na_2_SiO_3_. The measured adsorption of collector onto smithsonite was found to correlate well with flotation test results. It was concluded that hydrocarbon chains can be held accountable for the difference in the flotation performance with different amines. The longer the hydrocarbon chain, the stronger the hydrophobic association ability of amine, which is conducive to the selective amine adsorption onto sulfurized smithsonite particles and hence the smithsonite flotation.

## 1. Introduction

Wulagen Zn mine is located in Ulugqat County, Kashgar City, Xin Jiang, China. The zinc ore mainly contains sulfide mineral sphalerite and oxidized mineral smithsonite, with an average Zn grade of 2.5%. The main gangue minerals in the ore are quartz, calcite, dolomite, sericite and illite. Zinc in the Wulagen mine is recovered through a typical Zn flotation with conventional reagents, including xanthate, pine oil and CuSO_4_. Due to the difficulty in recovering smithsonite, nearly 30% zinc is lost to the flotation tailings, which have an average zinc grade of 0.5–1.2%. Apart from the poor floatability of smithsonite, the formation of slimy slurries by the abovementioned clayed gangue minerals also accounts for poor zinc flotation performance.

The recovery of base metal oxide minerals by flotation with various reagents has been extensively studied. Carboxylic and hydroxamic acids [1] can be utilized to float zinc oxide mineral directly without the sulfidization process. The literature indicates that these collectors have performed well in the recovery of zinc oxide mineral [2], particularly hydroxamic acids that are more collective due to their strong chelating ability [3]. It should be noted, however, that these anionic collectors are generally limited to their selectivity, especially when processing oxide ores with a high content of carbonate gangue minerals [4]. It is well known that carbonate gangue minerals, such as calcite and dolomite, tend to be floated readily by these collectors.

Another method is to sulfurize the oxide mineral with Na_2_S or NaHS, followed by the flotation using xanthate, which is particularly commonly seen in the flotation of copper and lead oxide minerals [5]. However, industrial practice has shown that most xanthates are not fit for recovering zinc oxide minerals despite their sulfidization. The short carbon chain length of most commercially used xanthates (carbon atomic numbers of 2–5) may account for their poor collectivity [6,7,8]. 

Amines are also often used in the flotation of sulfurized zinc oxide minerals. For example, in the flotation of smithsonite, regular fatty amines have proven to be collective, but the poor selectivity limits their application only to high-grade zinc oxide ores [9]. It has been reported that zinc oxide ores that have at least a grade of over 5% Zn can be well floated with amines [10,11]. Of course, the flotation performance is also strongly associated with the presence of clayed mineral slimes. Interestingly, a recent study observed that amines could also be both collective and selective in the flotation of zinc oxide ores when pH reached 12 or higher [12].

In this study, a flotation tailing with a zinc grade less than 1% in the Wulagen zinc mine was processed via laboratory flotation with different combinations of Na_2_S and cationic amines. The tested amines include three ether amines with different molecular structures and a fatty amine, dodecyl amine. The effects of these amines on the zinc recovery were investigated, and their interactions with other reagents were also identified, with an aim of screening out a proper reagent scheme to improve zinc separation from extremely low-grade zinc flotation tailings.

## 2. Experimental

### 2.1. Ore Sample

The ore samples for the flotation tests were prepared using the Wulagen plant tail slurry. The slurry was allowed to settle for 24 h. With the supernatant decanted, the settled tailings were air dried, divided into 400 g by coning and quartering and packed for use. The samples showed an average zinc grade of 0.88%, and this zinc was in the main form of smithsonite (see Figure 1). Figure 2 shows the particle size distribution of the ore sample obtained by wet sieving.

### 2.2. Reagents

In this study, industrial-grade sodium sulfide (minimum active content > 60%, Shandong Guanghui Chemicals, Shandong, China) was used as the sulfidizing agent. For smithsonite sulfidization, a 10 wt.% Na_2_S solution was prepared, within four hours prior to flotation.

Table 1 shows the amines used for preparing collectors in the laboratory flotation tests. The amines include a fatty amine, dodecyl amine (DDA), and three ether amines with different molecular structures, i.e., OPA 10, OPA1214 and OPA 13. These amines were obtained from Xin Guang Chemistry Co., Ltd., Shandong, China, with a purity of > 98%, and kept in nitrogen-filled air-tight bags to prevent reaction of amine with CO_2_ before use.

The collector required for each flotation test was prepared by mixing a type of the amines listed in Table 1 with hydrochloric acid purchased from Sinopharm at a molar ratio of 4:1. This mixture was then diluted to a solution of 1 wt.% amine concentration using Xuzhou tap water, followed by a 10 min vigorous shake to ensure a sufficient and even mixing. Note that all the ether amine solutions (1 wt.%) were transparent, colorless and homogeneous, whereas DDA tended to precipitate in water, especially at low temperatures, so a slightly opaque DDA solution (1 wt.%) could be observed even in the homogenous state.

Industrial-grade sodium silicate (sodium metasilicate pentahydrate Na_2_O·nSiO_2_·5H_2_O, 2.6 < n < 2.8), obtained from Qingdao Haiwan Chemicals, China, was used as the dispersant as well as the inhibitor for quartz. Sodium carbonate (purity > 99.2%), obtained from Sinopharm holdings Xuzhou Co., Ltd., China, was used as the pH modifier. These reagents were prepared daily to a 10 wt.% solution using Xuzhou tap water before use.

Note that all reagent concentrations were given without considering the active content or crystalline water in the calculation, which is a common method adopted by flotation operators in ore dressing plants to add reagents into flotation systems.

### 2.3. Batch Flotation Tests

The flotation tests were carried out in a 1 L XFD batch flotation cell (Wuhan Mining Lab Equipment Company, Hubei, China) in which air was self-aspirated by an impeller, and the air flowrate was controlled by a valve. The feed slurry with a density of 35% was prepared by mixing the 400 g of ore sample and 743 g of water in the cell. Agitation rate was set to 2000 rpm. The reagents added to the cell were in the exact following sequence: pH modifier first, then followed by depressant and collector. The pH of the pulp was adjusted to 9. Na_2_S was conditioned for 3 min, and the other reagents were conditioned for 2 min. When flotation started, the air flowrate was controlled at 3.3 L/min during froth collection. The total flotation time was 5 min, and the froth was scraped every 8 s. At the completion of flotation, the concentrate and the tailing products were filtered, dried, weighed and assayed as well as analyzed for composition with X-ray fluorescence (XRF).

For the test in which desliming before flotation was required, the slurry was agitated in the cell for at least 3 min, then allowed to settle for 5 min, and the supernatant was syphoned out, filtrated and dried for assay. Xuzhou tap water was added to the flotation cell to compensate for the removed water. The desliming amount (%) was obtained from dividing the mass of fine/ultrafine particles removed from the sample by the total mass of the original sample.

### 2.4. Adsorption Tests

The adsorption of collector onto pure smithsonite particles was investigated to justify the differences in the zinc flotation with different collectors. Pure smithsonite particles were obtained from Wulagen sulfide ore tailings by manual picking using a microscope, and the selected particles were ground to less than 2 μm with an agate mortar. After splitting, 1 g of the ground smithsonite was transferred to a beaker, followed by the addition of 15 mL of deionized water into the beaker. Then, reagents were added as required for each adsorption test, simulating the conditions of the flotation tests. The slurry was magnetically stirred for 2 h. After standing and layering, the supernatant of the upper layer was taken for the measurement of residual collector concentration via eosin Y spectrophotometry. Thus, the adsorption amount of the collector onto smithsonite particles can be calculated by the difference between the initial amount of the collector and the residual amount of the collector in the liquid.

## 3. Results and Discussion

### 3.1. Effect of Collector

The beneficiation of ultrafine particles through flotation is difficult, especially in the presence of clayed gangue minerals [13,14]. Whether desliming or not also greatly affects the reagent scheme in a flotation [15]. Table 2 shows the zinc grade and zinc distribution of the sample on a size-by-size basis (wet sieving) that were obtained using inductively coupled plasma–atomic emission spectrometry (ICP-AES). Two replicates of the measurements were performed, and in Table 2, the average values were used, with a standard deviation being less than 5%. The total zinc grade was the weighted average of the zinc grade of each size fraction, which had no discernible difference from the measured one. For the ultrafine size fraction (−15 μm), the zinc grade was below the average, and the weight percentage was only 11.85%, suggesting that the removal of the fine mud would highly probably be beneficial to zinc grade in the concentrate without affecting the recovery significantly.

Figure 3 shows the effects of desliming on the concentrate zinc grade and zinc recovery. The flotation was conducted at the condition of 1500 g/t Na_2_CO_3_, 300 g/t Na_2_SiO_3_, 6000 g/t Na_2_S and 50 g/t collector. The error bars represent the standard deviation obtained from three replicate flotation tests. Apparently, desliming prior to flotation was found to result in a higher concentrate grade and recovery compared with non-desliming. In general, with an increase in the desliming amount, the concentrate grade increased gradually, regardless of the collectors used. The zinc recovery showed an evident increase first and then decreased when the desliming amount reached 10 wt.%. This indicates that a proper desliming was beneficial to improve the flotation of smithsonite, but excessive desliming reduced the flotation recovery. A better desliming amount in this study was found to be about 10 wt.%.

As also indicated in Figure 3, the effect of desliming on the smithsonite flotation performance was subject to the amines used in the flotation. Over the collectors, OPA 1214 greatly advanced the flotation recovery, the concentrate grade and the flotation kinetics when desliming prior to flotation was adopted. Though an approximate 10 wt.% of particles (i.e., the ultrafine and fine slime) was removed, the recovery using OPA 1214 increased by over 25% compared to that without desliming.

The effect of the collector on the flotation performance was further investigated by varying the collector concentration (ranging from 25 to 125 g/t) at the condition of 10 wt.% desliming amount, 1500 g/t Na_2_CO_3_, 300 g/t Na_2_SiO_3_ and 6000 g/t Na_2_S. Figure 4 shows the changes in the concentrate grade and recovery as a function of collector concentration. An increase in the collector concentration was found to result in an increase in the zinc recovery but a decrease in the concentrate zinc grade. Presumably, there was an increase in both the entrainment recovery of gangue mineral particles and the flotation recovery of gangue-interlocked valuable minerals when increasing the collector concentration. DDA exhibited the poorest flotation performance, while OPA 1214 showed the best, indicating the superiority of OPA 1214 to the other three collectors in both the collectivity and selectivity. A better grade and recovery rate were obtained when the collector concentration reached 50 g/t.

### 3.2. Interactions of Collector with Other Reagents

It is the combination of collector with other flotation reagents (e.g., sulfurizing agents and dispersants) that determines the overall smithsonite flotation performance, and the type of collector is of great importance for an economic reagent scheme design due to reagent interactions [16,17,18,19,20]. Figure 5 shows the interactive effects of the four amines and sodium sulfide on the flotation performance. By increasing the concentration of sodium sulfide, concentrate zinc grade and recovery increased first and then decreased. These trends agree with the observations made by Mehdilo et al. [16] that both zinc mineral recovery and grades were promoted by sodium sulfide at a low dosage, while depression of zinc flotation occurred when sodium sulfide was overdosed. To achieve the same concentrate zinc grade or zinc recovery, the sodium sulfide concentration was different for different amines. Compared with DDA, a relatively low amount of sodium sulfide was used in the smithsonite flotation with the three ether amines, among which OPA 1214 out-performed in saving the usage of sodium sulfide. Of course, the degree of the usage saving was also dependent on the selected values of the flotation indicators, such as concentrate zinc grade and recovery.

Figure 6 shows the interactive effects of the four amines and sodium silicate on the flotation performance. An increase in sodium silicate concentration increased the concentrate zinc grade rapidly without exerting a detrimental effect on Zn recovery when the sodium silicate concentration was below 500 g/t. However, both the concentrate zinc grade and recovery deteriorated at a higher sodium silicate concentration. Similar to sodium sulfide, different amounts of sodium silicate were required to reach the same flotation zinc recovery or concentrate zinc grade in the smithsonite flotation with different amine collectors. OPA 1214 also outperformed the other collectors in saving the usage of the depressant.

### 3.3. Mechanisms

The mechanism behind the effects of the collectors on the flotation performance was explored through the adsorption tests.Figure 7 shows the amount that each collector adsorbed onto smithsonite particles at different collector concentrations in the presence of 500 g/t sodium silicate and 6000 g/t sodium sulfide. In the range of the tested collector concentrations, increasing the concentration enhanced the adsorption of the collectors onto the particles. The ether amines in the adsorption amount in decreasing order were as follows: OPA 1214, OPA 13 and OPA 10, which showed a great advantage over the fatty amine DDA in the adsorption. As can be seen, the adsorption test results are in line with the flotation test results as shown in Figure 4.

Figure 8 and Figure 9 show the adsorption of the four collectors onto smithsonite particles at different sodium sulfide and sodium silicate concentrations, respectively. The results of the adsorption tests also correlated well with the flotation test results outlined in Figure 5 and Figure 6. Similarly, under these circumstances, the collectors were ranked as OPA 1214, OPA 13, OPA 10 and DDA in a decreasing order based on the adsorption onto smithsonite particles.

Based on the experimental results presented above, a possible mechanism can be derived for the flotation of smithsonite with ether amines (see Figure 10). It is known that surface potential of smithsonite particles is negative at a pH of 9 [5,21]. When separating smithsonite from tailings in the presence of sodium carbonate and sodium sulfide at the pH of 9, there is an improvement in both the hydrophobicity of smithsonite particle surface and the hydrophilicity of gangue minerals. On the one hand, the smithsonite particle surface becomes less hydrophilic with adsorption of HS^-^ and S^2-^ in the presence of chemisorbed sulfide ions, which improves the surficial hydrophobicity to some extent. On the other hand, gangue minerals such as quartz become more hydrophilic in the presence of sodium silicate, as a hydrophilic membrane can be formed on gangue mineral particle surfaces.

Amines mainly exist in the suspension in the form of RNH_2_(aq) at the pH of 9, and RNH_2_ may attach to the zinc in the form of ZnS through complexation bonds on the smithsonite particle surface as shown in Figure 11 [22]. The experimental results reported in this study indicate that it is the hydrocarbon chain in the amine molecular structure that greatly affects the adsorption of the amines onto smithsonite particles and thus the smithsonite flotation. Of the tested four amines, it was found that a longer hydrocarbon chain favors the complexation between amines and ZnS, and amines with straight hydrocarbon chains were found to perform better than branched hydrocarbon chains. It was reported that increasing a hydrocarbon chain length could affect the electron cloud density of nitrogen atoms in amines [23,24], and this inductive effect becomes greater with an increase in the number of carbon atoms. Thus, the long hydrocarbon chain could enhance the inductive effect to improve the collecting ability of the amine group. Additionally, the hydrophobic association energy between hydrocarbon chains is proportional to the length of the carbon chain. The longer the carbon chain, the stronger the hydrophobicity of the mineral surface. Thus, it was speculated that OPA 1214 can produce more hydrophobic sites than short-chain OPA13, OPA10 and DDA (see Figure 10). This suggests that smithsonite flotation can be improved with an appropriate structure of amines.

## 4. Conclusions 

Different combinations of amines and Na_2_S resulted in different concentrate zinc recovery and grade in the flotation of Wulagen sulfide flotation tailings (<1% Zn). DDA exhibited the poorest flotation performance, while OPA 1214 showed the best, indicating the superiority of OPA 1214 to the other three collectors in both the collectivity and selectivity. Increasing the concentration of each collector could improve the zinc recovery but reduce the concentrate zinc grade. There were interactions observed between different amines and Na_2_S and Na_2_SiO_3_, and OPA 1214 outperformed the others in saving the usage of both the Na_2_S and Na_2_SiO_3_. The adsorption test results correlate well with the flotation test results. It was concluded that the hydrocarbon chain was accountable for the difference in the flotation performance with different amines. The hydrocarbon chain could affect the electron cloud density of nitrogen atoms in amines, and the longer the hydrocarbon chain, the stronger the hydrophobic association ability of amines, which favors the selective amine adsorption onto sulfurized smithsonite particles and hence the smithsonite flotation.

## Figures and Tables

**Figure 1 molecules-26-05365-f001:**
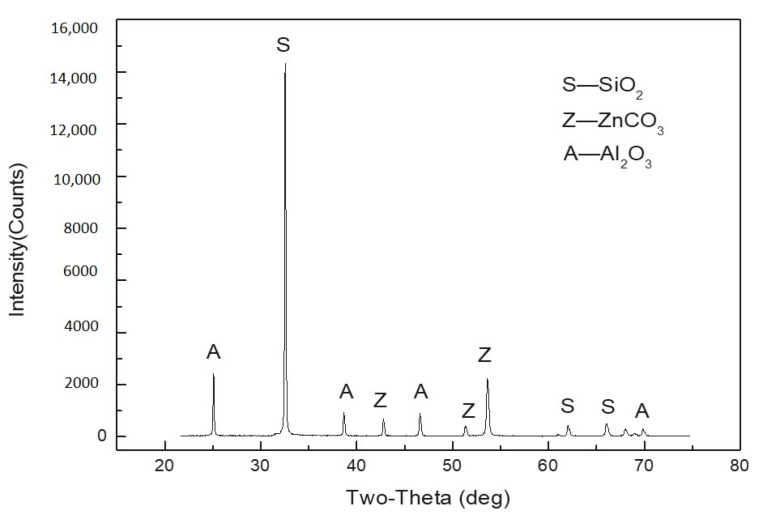
X-ray diffraction pattern of the feed sample.

**Figure 2 molecules-26-05365-f002:**
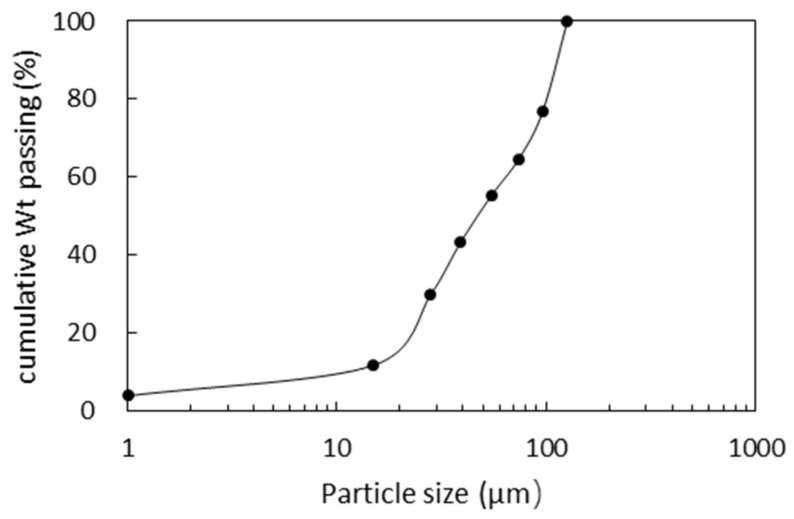
Particle size distribution of the feed sample obtained by wet sieving.

**Figure 3 molecules-26-05365-f003:**
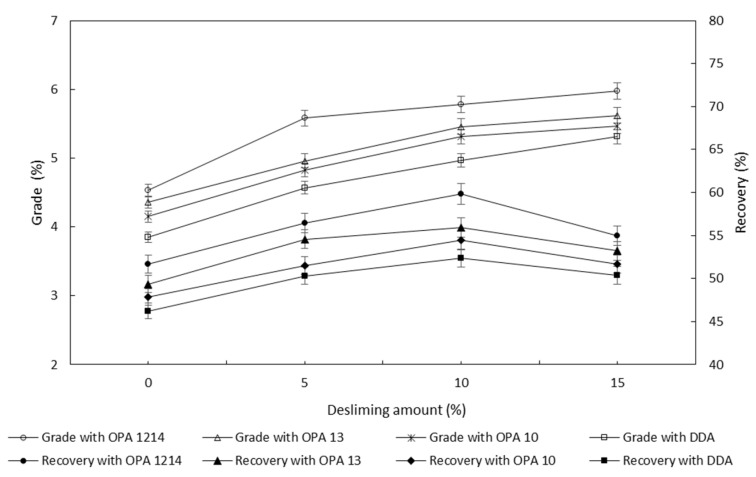
Concentrate Zn grade and recovery as a function of desliming amount.

**Figure 4 molecules-26-05365-f004:**
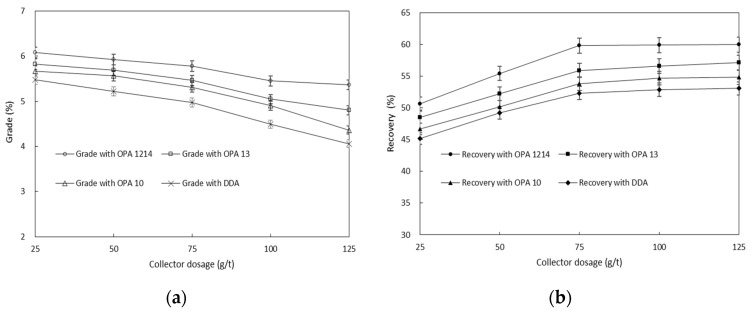
Effect of collector concentration on the concentrate Zn grade (**a**) and recovery (**b**).

**Figure 5 molecules-26-05365-f005:**
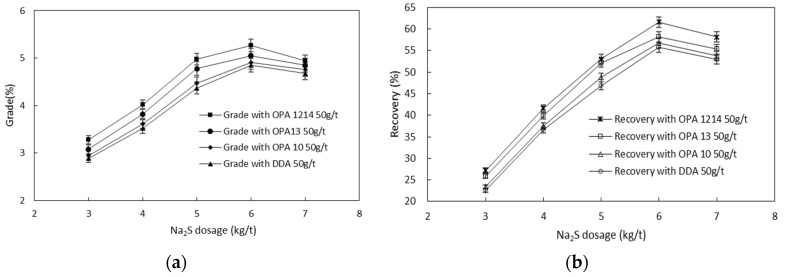
Interactive effect of Na_2_S and the four amines on the concentrate Zn grade (**a**) and recovery (**b**) in smithsonite flotation.

**Figure 6 molecules-26-05365-f006:**
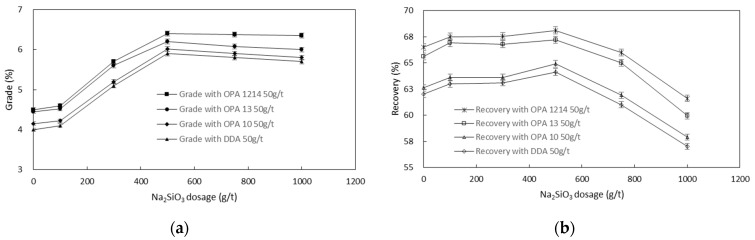
Interactive effect of Na_2_SiO_3_ and the four amines on the concentrate Zn grade (**a**) and recovery (**b**) in smithsonite flotation.

**Figure 7 molecules-26-05365-f007:**
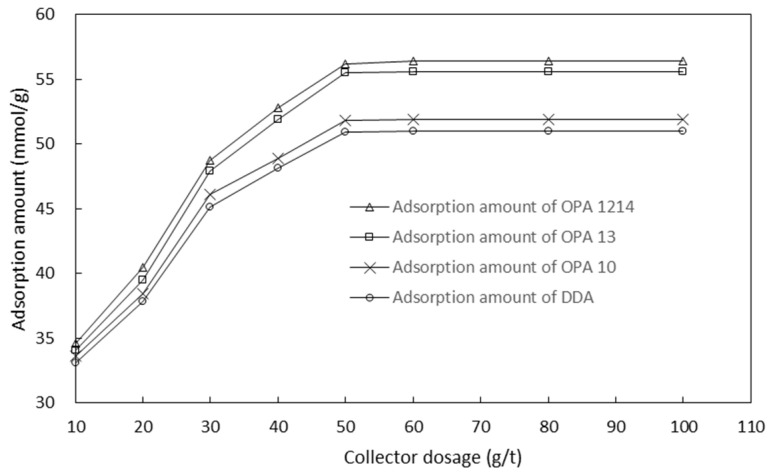
Adsorption amount of each collector on smithsonite particles at different collector concentrations.

**Figure 8 molecules-26-05365-f008:**
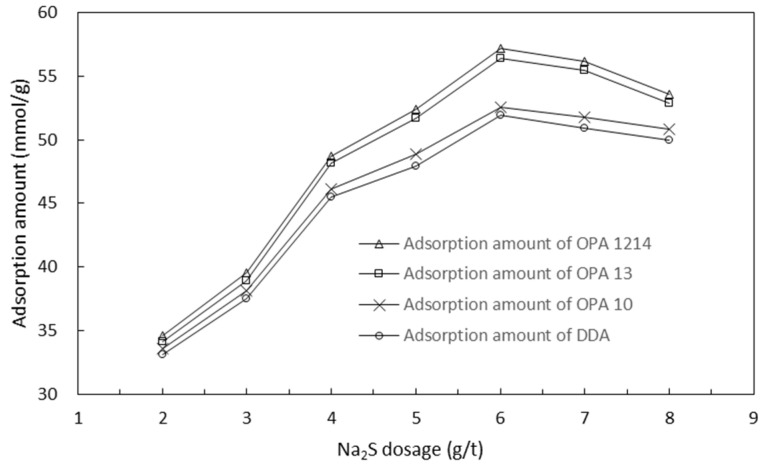
Adsorption amount of each collector on smithsonite particles at different Na_2_S concentrations (Na_2_SiO_3_ of 500 g/t).

**Figure 9 molecules-26-05365-f009:**
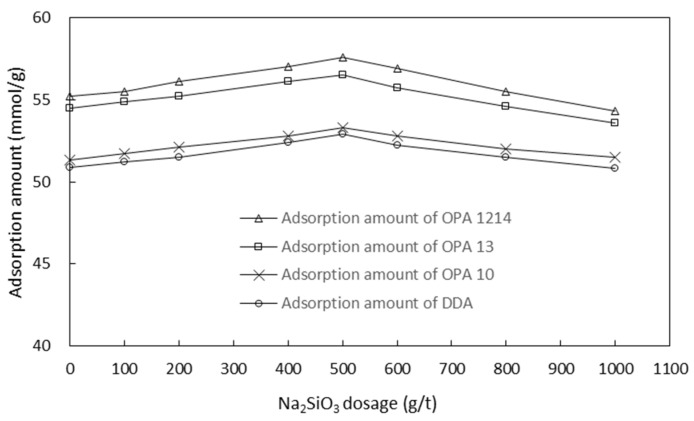
Adsorption amount of each collector on smithsonite particles at different Na_2_SiO_3_ concentrations (Na_2_S of 6000 g/t).

**Figure 10 molecules-26-05365-f010:**
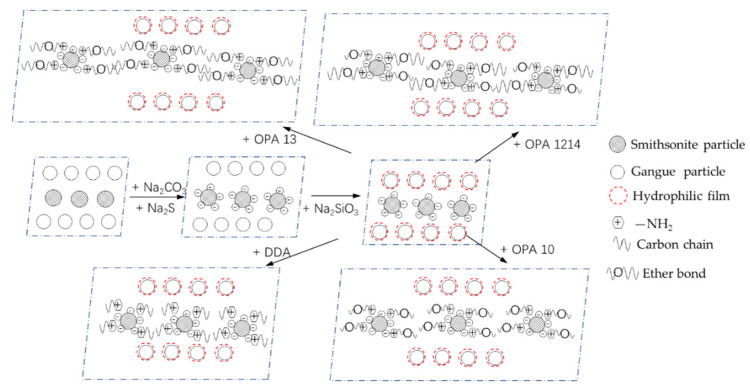
Schematic of mechanisms by which four amines affect smithsonite flotation.

**Figure 11 molecules-26-05365-f011:**
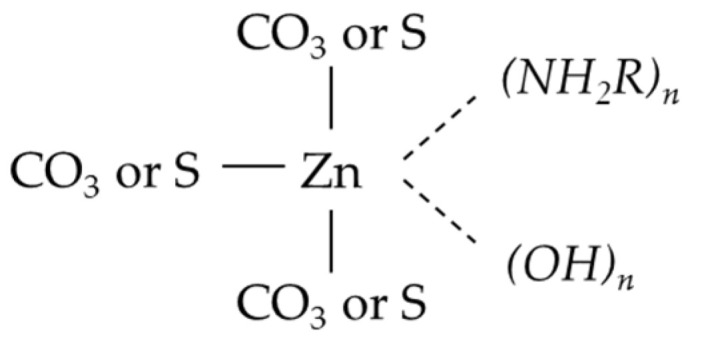
Complexation of amines and ZnS on smithsonite particles [22].

**Table 1 molecules-26-05365-t001:** Summary of amines used for preparing collectors in batch flotation tests.

Name	Abbreviation	Chemical Formula	Total Amine Value (mg KOH/g)	Distribution of Carbon Chain (%)
Normal dodecyl amine	DDA	CH_3_(CH_2_)_11_NH_2_	285–305	C_12_ ≥ 98, C_14_ ≥ 1, C_10_ ≥ 1
3-(iso-decyloxy)-1-propyl amine	OPA 10	(CH_3_)_2_CH(CH_2_)_7_-O-(CH_2_)_3_NH_2_	≥235	C_10_ ≥ 98C_8_ ≤ 2
3-(normal dodecyloxy)-1-propyl amine	OPA 1214	CH_3_(CH_2_)_11_-O-(CH_2_)_3_NH_2_	≥205	30 ≥ C_12_ ≥ 22
3-(normal tetradecyloxy)-1-propyl amine		CH_3_(CH2)_13_-O-(CH_2_)_3_NH_2_		78 ≥ C_14_ ≥ 68
3-(iso-tridecyloxy)-1-propyl amine	OPA 13	(CH_3_)_2_CH(CH_2_)_10_-O-(CH_2_)_3_NH_2_	≥200	C_13_ ≥ 98

**Table 2 molecules-26-05365-t002:** Zinc grade and distribution of feed sample.

Size (μm)	Wt. (%)	Zn Grade (%)	Zn Distribution (%)
+96	23.14	0.67	17.56
−96 + 74	12.11	0.75	10.29
−74 + 44	10.80	0.83	10.15
−44 + 37	10.49	1.03	12.24
−37 + 15	31.61	1.09	39.02
−15	11.85	0.80	10.74
Total	100.00	0.88	100.00

## Data Availability

The data presented in this study are available in article.

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
