# Peer review of "Zinc Recovery from Wulagen Sulfide Flotation Plant Tail by Applying Ether Amine Organic Collectors"

_molecules, 2021, doi:10.3390/molecules26175365_

Round 1
Reviewer 1 Report
The manuscript is well written written and contain original information that contribute to the ongoing efforts to recovery base metals from different waste streams. I recommend the publication of this article after completing the following revisions, with some of the comments in the dialog should not be ignored:
- please consider carefully to modify the title: could this be one option - "Zinc Recovery from Wulagen Sulfide Flotation Plant Tail by Applying Ether Amine Organic Collectors"
- detailed comments are attached as annotated PDF of the original manuscript, please consider also the comments given as a dialog.

Author Response
Dear Reviewer 1,
We appreciate your careful reading and valuable comments for improving the quality of this manuscript, and we have revised the manuscript based on your comments.
- Amend the title to "Zinc Recovery from Wulagen Sulfide Flotation Plant Tail by Applying Ether Amine Organic Collectors".
- Line 13, add the words "in China".
- Line 32, amend "2.50%" to "2.5%".
- Line 68, add the punctuation ".".
- Line 99, why do you use ≤ or ≥, almost all articles this days use < or >!
Response to 1-5: All these comments have been addressed in the revised manuscript.
- Amend "Zn distribution" in Table 2 to "Zn distributabtion".
Response: We have double checked this word, and are aware that “distribution” used in the original manuscript is not a typo.
- How these values were determined? And what is your error estimates (uncertainty) values? please include these information the Table caption or in the manuscript text indicating Table 2!
- WHAT does this 0.88 means in Table 2, which is given as total value for Zn grade (%)?! Please carefully consider this.
Response to 7-8: Based on your comments, we have double checked our experimental data and added the experimental description to this section in the revised manuscript to clear it up:
"Table 2 shows the zinc grade and zinc distribution of the sample on a size-by-size basis (wet sieving) that were obtained using inductively coupled plasma - atomic emission spectrometry (ICP-AES). Two replicates of the measurements were performed, and in Table 2 the average values were used, with a standard deviation being less than 5%. The total zinc grade was the weighted average of the zinc grade of each size fraction, which has no discernible difference with the measured one."
- Lines 266-267, concentrate zinc recovery and grade" is revised to "concentrations of zinc".
Response: We have checked this sentence carefully. We realized that "concentrate zinc recovery and grade" and "concentrations of zinc" may express the same thing, but in the field of mineral processing it would be more appropriate to use the term “recovery and grade of the flotation concentrate”. In the revised manuscript, therefore, it was decided to keep “concentrate zinc recovery and grade” in the sentence.
- Lines 270, add the word "regards".
Response: the word “regards” have been added to the sentence in the revised manuscript.

Reviewer 2 Report
The authors have extensively presented on the recovery of zinc from tailings obtained from a Sulfide flotation plant using Ether amine organic collectors. The experimental work have been described in details along with the results. The authors have appropriately cited other works and have made very good deductions based on the experimental work. The authors have duly explained the adsorption mechanisms which take place during flotation of smithsonite using the amine collectors. The article could be considered for publication, however there are some minor issues that the authors may consider before final publication.
- Line 172-173, the authors should consider explaining the their observation. What are the possible reasons which resulted in enhanced recovery at the expense of grade.
- Consider using the word "outperformed" or "performed better than" in place of "outdid"
Author Response
Dear Reviewer 2,
We appreciate your careful reading and valuable comments for improving the quality of this manuscript, and we have revised the manuscript based on your comments.
- Line 172-173, the authors should consider explaining their observation. What are the possible reasons which resulted in enhanced recovery at the expense of grade.
Response: In mineral flotation, it is often observed that flotation recovery increases while concentrate grade decreases. When increasing the collector dosage, more valuable mineral particles can be recovered due to the increased mineral surface hydrophobicity. This may also modify the surface properties of gangue-interlocked valuable minerals and thus improve the recovery of gangue-interlocked valuable minerals. Besides, the increase in the flotation recovery of valuable minerals will bring more water into the concentrate. In this case, more gangue particles that are dispersed in water also enter the concentrate with the water. This may well explain the observed increased recovery and reduced grade of the concentrate.
Based on this, we have revised the manuscript by providing a reasoning for this observation:
“Presumably there was an increase in both the entrainment recovery of gangue mineral particles and the flotation recovery of gangue-interlocked valuable minerals when increasing the collector concentration.”
- Consider using the word or "performed better than" in place of "outdid".
Response: This has been addressed in the revised manuscript.
